# Higher *Bifidobacterium* spp. fecal abundance is associated with a lower prevalence of hyperglycemia and cardiovascular risk markers among schoolchildren from Bahia, Brazil

**Camilla A. Menezes**[1], **Dalila L. Zanette**[2], **Letícia B. Magalhães**[1], **Jacqueline Tereza da Silva**[3], **Renata M. R. S. Lago**[4], **Alexvon N. Gomes**[4], **Ronald A. dos Santos**[1], **Ana Marice T. Ladeia**[4], **Nelzair A. Vianna**[1], **Ricardo R. Oliveira**[1]*

1 Gonçalo Moniz Institute, Oswaldo Cruz Foundation, Fiocruz, Salvador, Bahia, Brazil, 2 Carlos Chagas Institute, Oswaldo Cruz Foundation, Fiocruz, Curitiba, Paraná, Brazil, 3 Global Academy of Agriculture and Food Systems, The University of Edinburgh, Edinburgh, Scotland, United Kingdom, 4 Bahiana School of Medicine and Public Health, Salvador, Bahia, Brazil

* ricardo.riccio@fiocruz.br

**Data Availability Statement:** The dataset used to support the results presented in this manuscript is

## Abstract

The gut microbiome has recently been the subject of considerable scientific interest due to its essential bodily functions. Several factors can change the composition and function of the gut microbiome, and dietary habits are one of the most important contributors. Despite the recognition of the probiotic effects related to the genus *Bifidobacterium* spp. (BIF) studies aiming to assess its relationship with metabolic outcomes show conflicting results, particularly in the child population. This cross-sectional study aimed to evaluate the fecal abundance of BIF in a group of schoolchildren from public schools in Bahia, Brazil, and to investigate their relationship with food consumption and laboratory and anthropometric characteristics. A sample of 190 subjects aged 5 to 19y was randomly selected for dietary, laboratory, and anthropometric assessment. Fecal BIF abundance assessment was performed using the Real-Time Polymerase Chain Reaction assay. Fecal BIF abundance was higher among subjects who had lower intakes of meat. The abundance of BIF was also higher among subjects with lower Waist Circumference and Waist-to-Height Ratio (WHtR). Low BIF abundance was associated with a higher prevalence of hyperglycemia (PR 1.04, 95%CI 1.02–1.07, p = 0.001) and high WHtR (PR 1.04, 95%CI 1.01–1, 08, p = 0.015). These findings allow us to conclude that BIF fecal abundance is related to dietary and anthropometric parameters in schoolchildren, and its increase is associated with positive metabolic outcomes.

## Introduction

The human gut microbiome has been the subject of considerable scientific interest in recent years, particularly after the development of metagenomic studies [1]. Microbiota refers to

available at the FIOCRUZ institutional data repository ARCA DADOS (https://arcadados. fiocruz.br/dataset.xhtml?persistentId=doi:10. 35078/QO6VZB).

**Funding:** C.A.M. receives a doctorate research grant from the Research Support Foundation of the State of Bahia (https://www.fapesb.ba.gov.br/) BOL0170/2019. The founders had no role in study design, data collection and analysis, decision to publish, or preparation of the manuscript.

**Competing interests:** The authors have declared that no competing interests exist.

microorganisms, including bacteria, fungi, viruses, and some unicellular eukaryotes [2]. The digestive microbiota is the complex community of microorganisms living in humans' and animals' gastrointestinal tracts, including insects [3]. In humans, the intestinal microbiota concentrates most microorganisms and the most significant number of species compared to other body parts, with the largest concentration being in the large intestine [4]. The genera *Bacteroides* spp., *Bifidobacterium* spp., and *Lactobacillus* spp. are the most prevalent, which suggests that they are particularly significant for the host organism physiology [5].

A healthy gut microbiome develops essential bodily functions, which can be classified into metabolism, protection, and tropism [2]. The metabolism function is related to the synthesis, digestion, and absorption of nutrients [6] and the modulation of gut-brain communication [7, 8]. The protection function is associated with preventing pathogenic microorganisms' growth, either by colonization site competition or by the ability to produce antimicrobial peptides [9]. Finally, the trophic position occurs by stimulating the proliferation and differentiation of the intestinal epithelium and developing and modulating the immune system [10, 11].

The genus *Bifidobacterium* spp. belongs to the phylum *Actinobacteria* and comprises more than 50 Gram-positive species [12]. Probiotic effects related to this genus include the recovery of the intestinal microbiota after antimicrobial therapy [13], reduction of serum cholesterol levels through the degradation and absorption of bile acids [14], and immunomodulatory activity [15]. Despite this, studies that aim to assess the relationship between *Bifidobacterium* spp. and metabolic outcomes, such as obesity and cardiovascular disease, present conflicting results [16, 17], especially in the child population [18, 19].

In addition, Brazil follows the global trend of reducing the prevalence of underweight and increasing overweight and obesity in the school-age population, currently recognized as a public health problem [20]. Obesity is associated with the development of Chronic Noncommunicable Diseases (NCDs), which can develop because of multiple causes, with an unhealthy diet being one of the main modifiable risk factors [21]. Although clinical manifestations are more frequently observed in adulthood, exposure to risk factors has occurred at an increasingly early age, especially those related to diet [22].

Several factors can alter the composition and function of the intestinal microbiota [23]. In addition to genetic susceptibility, type of delivery, breastfeeding, age, geographic location, antibiotic use, and dietary pattern play an important role [24–28]. Therefore, this study aimed to evaluate the fecal abundance of *Bifidobacterium* spp. of a group of students from public schools in Bahia, Brazil, and to investigate its relationship with food consumption and with laboratory and anthropometric characteristics. It is hoped that the results of this study will contribute to the knowledge of the school-age population's health conditions at the local level and may guide the implementation and realignment of public policies on Food and Nutrition Security.

## Materials and methods

### Population and data collection

The population of this cross-sectional study is composed of students from Barrocas, Biritinga, Serrinha, Teofilândia, and Valente, in the interior of Bahia, Brazil, regularly enrolled in the municipal public education system in the years 2019 and 2020. Because it represents most of the school-age population, individuals aged 5 and 19 years were eligible and invited to participate in the study. Due to the potential to impact the gut microbiota, those with a previous diagnosis of food allergies and intolerances and those who used antibiotics 30 days before the fecal material collection were excluded. Considering an estimated overweight proportion of 15% (95% CI) and a desired precision of 5%, we obtained a sample of 190 randomly selected individuals.

A trained team collected clinical-demographic, dietary, laboratory, and anthropometric information in the morning at the school where the subjects were regularly enrolled.

## Clinical and demographic assessment

A face-to-face interview was guided by a previously structured questionnaire containing questions about the individual's clinical history, to be answered by the legal guardian. In addition to identification data, the questions included type of delivery; presence and duration of exclusive breastfeeding; disease history; family history; and medication use.

## Food consumption evaluation

The 24-hour recall was used to identify and quantify all foods ingested on the day before the interview. The 24-hour recall was conducted by the nutritionists of the research group, who were trained for this task. These nutritionists had prior access to the menus served in schools, which was crucial in understanding the local dietary habits. In addition, a photographic album of food portions was used to help fill in the information and more accurately determine the portioning of consumed food [29]. The images were accompanied by codes that were used to convert the portions into standardized measurement units (grams or milliliters). The consumption data of each participant were imputed into an Excel spreadsheet. Using R software, the Brazilian Food Composition Table [30] database was cross-referenced to identify the number of calories, macronutrients, and micronutrients consumed by each participant. For foods not available in that table, the Centesimal Food Composition Table of the Brazilian Institute of Geography and Statistics [31] was used. For qualitative evaluation, the NOVA classification was used [32].

## Blood and stool assessment

Blood samples were collected in an 8-hour fasting state by a specialized technical team and analyzed by the local Central Public Health Laboratory and local private laboratories, under the support and responsibility of the Health Department of the municipalities involved. The following indicators were evaluated: fasting glucose, total cholesterol, and fractions, triglycerides.

The participants collected stool samples for parasitological and microbiota analysis and placed them in sterile containers previously provided by the research team. The stool samples were transported at 4° C to local support laboratories, where an aliquot was extracted, which was later transported on dry ice to the Laboratory of Experimental Pathology at the Gonçalo Moniz Institute, Oswaldo Cruz Foundation, in Salvador, Bahia, Brazil, to be stored at -20° C until analysis. The remaining stool sample was sent for parasitological analysis in a private laboratory in the same city.

Stool samples bacterial DNA extraction was performed using the QIAamp PowerFecal DNA Kit® (QIAGEN, Canada), according to the manufacturer's specifications. Quantification and analysis of the purity of the extracted DNA were performed using the NanoDrop® spectrophotometry equipment (Thermo Fisher Scientific, United States). Interest microbiota quantification was performed by the Real-Time Polymerase Chain Reaction (RT-PCR) method, using the Real-time PCR 7500® equipment (Thermo Fisher Scientific, United States). For quantification of total bacteria (TB), Primer Forward `ACTCCTACGGGAGGCAGCAG` and Primer Reverse `ATTACCGCGGCTGCTGG` were used. For analysis of *Bifidobacterium* spp. (BIF) Primer Forward `GCGTGCTTAACACATGCAAGTC` and Primer Reverse `CACCCTTTCCAGGAGCTATT` were used (Ludwig Biotechnology®). Analyzes were performed using 1 μL of the extracted DNA, 5 μL of Sybr Green Master Mix® (Applied

Biosystems, United States), 0.1 μL of Primer Forward, 0.1 μL of Primer Reverse, and 3.8 μL of ultrapure water, adding 10 μL in the final reaction. The process was performed for both TB and BIF quantification. The amplification program conducted on the equipment was: 50˚ C for 2 minutes, 95˚ C for 10 minutes, followed by 40 cycles at 95˚ C for 1 second and at 60˚ C for 1 minute, adding the Melt Curve Stage (95˚ C for 15 seconds, 60˚ C for 1 minute, 95˚ C for 30 seconds, 60˚ C for 15 seconds).

The result of the target DNA amplification in RT-PCR is provided in a Threshold Cycle (Ct) value. In addition, a value of delta Ct (ΔCt) was extracted from it, the result of subtracting the Ct of the target gene (BIF) from the Ct of the reference gene (TB), as shown in Eq 1. The quantification results were presented in Relative Expression Units (REU), dividing 10,000 by 2 to the delta Ct, according to the model previously described [33], as shown in Eq 2. These results express an idea of *Bifidobacterium* spp. abundance about the total number of bacteria in the sample.

$$\Delta Ct = Ct \ of \ the \ target \ gene \ (BIF) - Ct \ of \ the \ reference \ gene \ (TB) \qquad (1)$$

$$REU = 10.000/2^{\Delta Ct} \qquad (2)$$

## Anthropometric evaluation

Nutritional status was classified by the Body Mass Index (BMI) for Age (BMI/A). With the student wearing the school uniform, weight was measured using a digital electronic scale (Seca®, Hamburg, Germany) with a maximum capacity of 150 kg and an accuracy of 0.1 kg. Height was measured with the student without shoes, using a portable vertical stadiometer (AVA-312®, Brazil) graduated in centimeters, with a maximum capacity of 2.10 m and accuracy of 0.001 m. BMI/A was classified according to the World Health Organization child growth curves [34]. Waist Circumference (WC) was assessed with an inelastic measuring tape (Balmak®, Brazil) with a measurement range from 0 to 150 cm and graduated in millimeters and classified according to the curves proposed by Fernández [35]. Waist-to-Height Ratio (WHtR) was calculated as suggested by McCarthy and Ashwell [36] and validated for children and adolescents by Nambiar and collaborators [37] to classify cardiovascular risk.

## Statistical analysis

Statistical analyzes were performed using R 4.1.0, GraphPad Prism 8.2.1, and Stata 11 software. To characterize the sample, a descriptive analysis was performed. After verifying the normality behavior of the numerical variables, using the Kolmogorov-Smirnov and Shapiro-Wilk tests, measures of central tendency and dispersion were established, considering the means and their respective Standard Deviation (SD) for the parametric variables, and the medians and their interquartile ranges (IQR) for the non-parametric ones. The Student's t-test for parametric variables and the Mann-Whitney test for non-parametric ones were used to compare these measures. The Kruskal-Wallis test was used to compare non-parametric variables between three or more groups. Categorical variables were compared using Pearson's chi-square test and Fisher's exact test when appropriate. Inferential statistics were performed using Pearson's correlation coefficient for parametric variables and Spearman correlation coefficient for non-parametric ones. The binomial logistic regression model was used to investigate association (Prevalence Ratio). P values lower than 0.05 were considered significant.

## Ethics statements

The research entitled "Evaluation of an intervention project in school meals on children and adolescents health in the interior of Bahia," in which this study is inserted was approved by the

Ethics Committee in Research on Human Beings of the Bahiana School of Medicine and Public Health on September 17, 2018, under protocol number 2962623, as determined by National Health Council resolution 466/2012 [38]. The participation was conditioned to voluntary agreement from the subject and their legal guardian, documented by signing the Term of Assent, when applicable, and the Term of Free and Informed Consent.

## Results

### Clinical and demographic assessment

Most of the sample was obtained from rural schools (57%) and was male (52%). The age ranged between 9 and 13 years (median 9.6 y, SD 2.8 y). Most of the population was born via vaginal delivery (64%), was exposed to breastfeeding (92%), had exclusive breastfeeding for at least the sixth month of life for 58% of them, and reported exposure to antibiotic therapy before five years of age (73%). Intestinal parasites investigation revealed infection by *Entamoeba histolytica* in 2 subjects and by *Giardia lamblia* in 3 participants. They were instructed about the treatment. In this population, the fecal abundance of *Bifidobacterium* spp. was not influenced by age, school location, birth delivery, breastfeeding, or antibiotic therapy in childhood (Table 1).

**Table 1. *Bifidobacterium* spp. fecal abundance according to demographic and clinical characteristics.**

| Indicators | *Bifidobacterium* spp. | | |
|---|---|---|---|
| | n (%) | REU[1] | p |
| **School location** | | | |
| Rural area | 109 (57) | 854 (200–1805) | 0.176[2] |
| Urban area | 81 (43) | 620 (104–2207) | |
| **Sex** | | | |
| Female | 91 (48) | 912 (188–2246) | 0.147[2] |
| Male | 99 (52) | 537 (110–1687) | |
| **Age** | | | |
| 5–8 y | 73 (38) | 447 (120–1762) | 0.099[3] |
| 9–13 y | 94 (49) | 1104 (188–2196) | |
| 13–19 y | 23 (13) | 443 (46–1915) | |
| **Birth delivery** | | | |
| Vaginal | 120 (64) | 797 (126–2176) | 0.936[2] |
| Cesarian | 68 (36) | 902 (139–1921) | |
| **Breastfeeding** | | | |
| Yes | 173 (92) | 791 (129–1768) | 0.369[2] |
| No | 15 (8) | 1001 (364–3036) | |
| **Duration of exclusive breastfeeding** | | | |
| >6 months | 100 (58) | 725 (132–2207) | 0.870[2] |
| <6 months | 73 (42) | 1001 (144–1756) | |
| **Use of antibiotics before five years of age** | | | |
| No | 49 (27) | 1111 (214–2230) | 0.272[2] |
| Yes | 132 (73) | 574 (118–1790) | |

[1] Median (interquartile range)

[2] Mann-Whitney test

[3] Kruskal-Wallis test.

p values in bold indicate statistically significant differences (< 0,05).

REU: Relative Expression Unit.

## Food consumption evaluation

To assess the fecal abundance of *Bifidobacterium* spp. according to the food consumption characteristics, each of the food items was categorized into subjects who consumed more (≥75th percentile) and those who consumed less (<75th percentile). *Bifidobacterium* spp. fecal abundance was lower among subjects with a higher meat intake (Table 2).

## Laboratory and anthropometric assessment

According to Table 3, there was no statistically significant difference in the fecal abundance of *Bifidobacterium* spp. between subjects who had hyperglycemia, hypercholesterolemia, and hypertriglyceridemia and those who did not. In terms of anthropometry, although there was no difference between nutritional status (BMI/A), the abundance of *Bifidobacterium* spp. was significantly higher among subjects with lower WC and WHtR when compared to those above the recommended value (909 IQR 153–2199 vs. 178 IQR 79–1135, p = 0.048; 902 IQR 145–2207 vs. 181 IQR 107–1026, p = 0.048, respectively).

In addition to the descriptive analysis, the relationship between the fecal abundance of *Bifidobacterium* spp. and the dietary, laboratory, and anthropometric characteristics were also investigated. Fig 1 presents Spearman's correlation graphs between fecal *Bifidobacterium* spp. abundance and possible health outcomes to investigate whether the abundance of this genus in the gut microbiota could behave as an exposure factor for laboratory and anthropometric findings. There was a very weak but statistically significant negative correlation (r = -0.180, p = 0.014) between the fecal abundance of *Bifidobacterium* spp. and the WHtR, demonstrating that the higher concentration of these bacteria was related to lower cardiovascular risk, using WHtR as a parameter for this classification.

Low *Bifidobacterium* spp. abundance was also associated with a higher prevalence of hyperglycemia (PR 1.04, 95%CI 1.02–1.07, p = 0.001). Likewise, the prevalence of high WHtR was 1.04 times higher in subjects who had a low *Bifidobacterium* spp. abundance when compared with those with higher concentrations (PR 1.04, 95%CI 1.01–1.08, p = 0.015). Data are shown in Table 4.

## Discussion

The gut microbiome of children born by vaginal delivery is more diverse and abundant in probiotic activity species than the microbiota of children born by cesarean delivery [26]. During the first days of life, *Escherichia coli*, *Clostridium* spp., and *Streptococci* spp. colonize the gastrointestinal tract, and during breastfeeding, *Bifidobacterium* spp. and *Lactobacillus* spp. arises [39]. The first year of life is the most critical period of gut microbiome development. After this phase, several factors influence the microbiota's quantity, diversity, and metabolism. The phylum *Actinobacteria* is predominant in children and adolescents, especially the genus *Bifidobacterium* spp. [40].

Children exclusively breastfed until the sixth month of life tend to show *Actinobacteria* growth and *Firmicutes* and *Proteobacteria* inhibition, resulting from the metabolism of oligosaccharides in human milk. Also, as human milk is a natural source of bifidobacteria, once breastfeeding is discontinued or complementary feeding is initiated, there is a reduction in the availability of this bacterial genus to colonize the intestine [26]. Simultaneously, the introduction of other foods, and therefore other energy substrates, promotes the growth of other bacterial genera. On the other hand, children fed with infant formula have a greater abundance of *Clostridium* spp. and *Bacteroides* spp. [40]. Individuals exposed to antibiotics use, especially up to the first five years of life, tend to show changes in the quantity, diversity, and metabolism of the microbiota due to the competitiveness mechanism change by which the microbiota

**Table 2.** *Bifidobacterium* spp. fecal abundance according to food consumption characteristics.

| Indicators | Food intake[1] | *Bifidobacterium* spp. | | |
|---|---|---|---|---|
| | | N | REU[1] | p[2] |
| **Energy (Kcal)** | 1755 (1309–2259) | | | |
| < 75th percentile | | 137 | 1058 (131–2286) | 0.091 |
| ≥ 75th percentile | | 46 | 531 (143–1165) | |
| **Protein (g/1000Kcal)** | 32,2 (27,4–39,4) | | | |
| < 75th percentile | | 137 | 791 (109–2192) | 0.867 |
| ≥ 75th percentile | | 46 | 819 (183–1437) | |
| **Carbohydrates (g/1000Kcal)** | 133,8 (115,0–151,4) | | | |
| < 75th percentile | | 139 | 791 (126–1856) | 0.520 |
| ≥ 75th percentile | | 44 | 965 (136–2173) | |
| **Fats (g/1000Kcal)** | 37,5 (31,6–44,3) | | | |
| < 75th percentile | | 137 | 902 (125–2169) | 0.862 |
| ≥ 75th percentile | | 46 | 754 (172–1614) | |
| **Saturated fatty acids (g/1000Kcal)** | 12,4 (9,9–15,4) | | | |
| < 75th percentile | | 138 | 574 (122–1827) | 0.142 |
| ≥ 75th percentile | | 45 | 1015 (290–2309) | |
| **Monounsaturated fatty acids (g/1000Kcal)** | 11,6 (9,6–14,1) | | | |
| < 75th percentile | | 138 | 817 (121–1872) | 0.446 |
| ≥ 75th percentile | | 45 | 797 (211–2301) | |
| **Polyunsaturated fatty acids (g/1000Kcal)** | 8,7 (6,4–11,0) | | | |
| < 75th percentile | | 138 | 909 (132–1827) | 0.739 |
| ≥ 75th percentile | | 45 | 529 (136–2326) | |
| ***Trans* fatty acids (g/1000Kcal)** | 1,4 (1,1–2,0) | | | |
| < 75th percentile | | 138 | 854 (137–1780) | 0.848 |
| ≥ 75th percentile | | 45 | 508 (126–2192) | |
| **Fiber (g/1000Kcal)** | 7,6 (6,2–9,0) | | | |
| < 75th percentile | | 137 | 769 (152–1756) | 0.631 |
| ≥ 75th percentile | | 46 | 1073 (66–3041) | |
| **Total sugar (g/1000Kcal)** | 48,9 (28,7–70,0) | | | |
| < 75th percentile | | 138 | 772 (151–1768) | 0.766 |
| ≥ 75th percentile | | 45 | 1303 (66–2441) | |
| **Unprocessed food (g/1000Kcal)** | 477,1 (354,3–597,7) | | | |
| < 75th percentile | | 137 | 797 (148–1768) | 0.580 |
| ≥ 75th percentile | | 46 | 888 (100–2973) | |
| **Unprocessed meat (g/1000Kcal)** | 93,0 (40,0–151,7) | | | |
| < 75th percentile | | 136 | 1088 (157–2681) | **0.007** |
| ≥ 75th percentile | | 47 | 430 (104–1088) | |
| **Unprocessed vegetables (g/1000Kcal)** | 245,0 (86,1–492,8) | | | |
| < 75th percentile | | 136 | 573 (135–1762) | 0.240 |
| ≥ 75th percentile | | 47 | 1134 (123–2365) | |
| **Processed food (g/1000Kcal)** | 38,4 (0,0–75,7) | | | |
| < 75th percentile | | 137 | 733 (117–2169) | 0.397 |
| ≥ 75th percentile | | 46 | 1022 (159–1780) | |
| **Ultra-processed food (g/1000Kcal)** | 61,5 (30,0–100,6) | | | |
| < 75th percentile | | 139 | 775 (126–1768) | 0.485 |

(*Continued*)

**Table 2.** (Continued)

| Indicators | Food intake[1] | *Bifidobacterium* spp. | | |
|---|---|---|---|---|
| | | N | REU[1] | p[2] |
| ≥ 75th percentile | | 44 | 1303 (152–2207) | |

[1] Median (interquartile range)

[2] Mann-Whitney test.

p values in bold indicate statistically significant differences (< 0,05).

REU: Relative Expression Unit.

"Unprocessed meat" was defined as any preparation with unprocessed red meat, poultry, pork, and fish. Any preparation with unprocessed fruits and vegetables was considered an "unprocessed vegetable." Processed foods are produced by adding culinary ingredients (salt, sugar, fats) to unprocessed foods, using preservation methods such as canning and bottling (e.g., canned beans, vegetables, fish, processed meat, and fruit jelly). Ultra-processed foods are industrial formulations composed entirely or mostly of substances extracted from food (oils, fats, sugar, proteins), derived from food constituents (hydrogenated fats, modified starch), and/or synthesized in a laboratory from raw materials and organic ingredients (colorants, flavorings, sweeteners, flavor enhancers) (e.g., sausages, soft drinks, ice cream, snacks).

**Table 3.** *Bifidobacterium* spp. fecal abundance according to laboratory and anthropometric characteristics.

| Indicators | Variable[1] | *Bifidobacterium* spp. | | |
|---|---|---|---|---|
| | | n (%) | REU[1] | p |
| **Fasting glucose (mg/dL)** | 85.0 (79.5–90.0) | | | |
| Desirable | | 172 (94) | 819 (140–2199) | 0.374[2] |
| High | | 11 (6) | 522 (89–1672) | |
| **Total cholesterol (mg/dL)** | 157 (136–180) | | | |
| Desirable/Tolerable | | 161 (88) | 866 (136–2192) | 0.376[2] |
| High | | 22 (12) | 465 (124–1329) | |
| **LDL cholesterol (mg/dL)** | 79 (68–102) | | | |
| Desirable | | 169 (92) | 775 (127–1888) | 0.473[2] |
| High | | 14 (8) | 1272 (144–3843) | |
| **Triglycerides (mg/dL)** | 72.5 (55.0–95.0) | | | |
| Desirable | | 152 (83) | 794 (127–1905) | 0.824[2] |
| High | | 31 (17) | 1058 (178–2161) | |
| **BMI (Kg/m²)** | 16.9 (15.0–19.6) | | | |
| Underweight | | 10 (5) | 358 (3–1112) | 0.212[3] |
| Eutrophy | | 112 (61) | 909 (132–2643) | |
| Overweight | | 62 (34) | 762 (155–1759) | |
| **WC (cm)** | 59.2 (54.0–67.0) | | | |
| Desirable | | 160 (86) | 909 (153–2199) | **0,048**[2] |
| High | | 25 (14) | 178 (79–1135) | |
| **WHtR** | 0.43 (0.40–0.46) | | | |
| Desirable | | 169 (91) | 902 (145–2207) | **0.048**[2] |
| High | | 16 (9) | 181 (107–1026) | |

[1] Median (interquartile range)

[2] Mann-Whitney test

[3] Kruskal-Wallis test.

p values in bold indicate statistically significant differences (< 0,05).

REU (Relative Expression Unit); BMI (Body Mass Index); WC (Waist Circumference); WHtR (Waist to Height Ratio).

It was considered as high blood glucose (≥ 100 mg/dL); high total cholesterol (> 200mg/dL); high LDL-cholesterol (≥ 130 mg/dL); high triglycerides (≥ 100 mg/dL for subjects aged 5 to 10 y, ≥ 150 mg/dL for subjects aged 11 to 19 y). Low weight was defined as the combination of thinness and severe thinness; overweight was defined as the combination of overweight, obesity, and severe obesity, according to the BMI for Age, according to the child growth curves of the World Health Organization [34]. The WC was classified according to the parameters established in the literature [35]. WHtR was considered high when ≥ 0.5 [36].

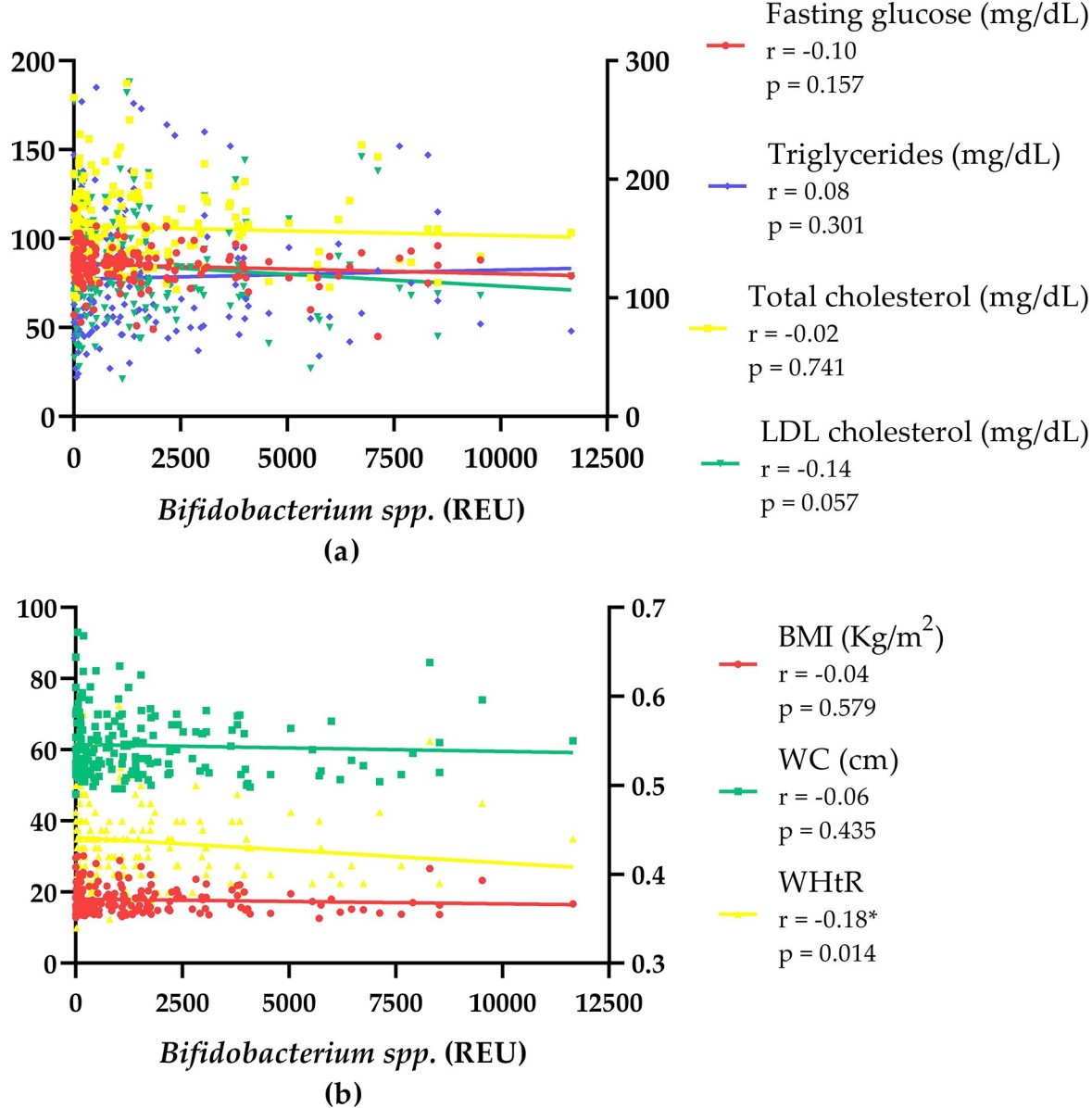

**Fig 1. Spearman correlation between *Bifidobacterium* spp. fecal abundance and possible health outcomes.** (a) Blood level parameters. Left Y-axis (Fasting glucose; Triglycerides; LDL-cholesterol); Right Y-axis (Total cholesterol). (b) Anthropometric parameters. Left Y-axis (BMI–Body Mass Index; WC–Waist Circumference); Right Y-axis (WHtR–Waist to Height Ratio). *Statistically significant difference (p < 0.05). REU: Relative Expression Unit. Spearman r classification–Very weak (0,00 to 0,19); Weak (0,20 to 0,39); Moderate (0,40 to 0,69); Strong (0,70 to 0,89); Very strong (0,90 to 1,00).

inhibits the colonization of pathogenic strains. This effect varies depending on the type of antibiotic, dose, and length of use [41].

In the population investigated in this study, there was no significant statistical difference in *Bifidobacterium* spp. fecal abundance depends on the delivery type, age, breastfeeding, and use of antibiotic therapy in childhood. It is worth noting that these variables do not represent information recorded in documents but reported by the legal guardians of the subjects, which implied an understanding of the questions asked at the time of the interview, in addition to relying on memory.

**Table 4. Association between *Bifidobacterium* spp. fecal abundance and health outcomes.**

| Indicators | *Bifidobacterium* spp. (REU) | | | |
|---|---|---|---|---|
| | Lowest concentration [1] | Highest concentration [1] | PR (95%CI) | p[2] |
| **Hyperglycemia** | | | | |
| No | 129 (75%) | 43 (25%) | 1 | |
| Yes | 11 (100%) | 0 (0%) | 1.04 (1,02–1,07) | **0.001** |
| **Hypercholesterolemia** | | | | |
| No | 130 (77%) | 39 (23%) | 1 | |
| Yes | 10 (71%) | 4 (29%) | 0.99 (0,94–1,04) | 0.664 |
| **Hypertriglyceridemia** | | | | |
| No | 115 (76%) | 37 (24%) | 1 | |
| Yes | 25 (81%) | 6 (19%) | 1.02 (0.96–1.09) | 0.528 |
| **Overweight** | | | | |
| No | 90 (74%) | 32 (26%) | 1 | |
| Yes | 50 (81%) | 12 (19%) | 1.05 (0.96–1.15) | 0.278 |
| **Cardiovascular risk** | | | | |
| No | 126 (74%) | 43 (26%) | 1 | |
| Yes | 15 (94%) | 1 (6%) | 1.04 (1.01–1.08) | **0.015** |

[1] n (%)

[2] Binomial logistic regression model.

p values in bold indicate statistically significant differences ($< 0,05$).

REU: Relative Expression Unit; PR: Prevalence Ratio; CI: 95% Confidence Interval.

REU of *Bifidobacterium* spp. above the 75th percentile were considered "Highest concentration," and values below the 75th percentile were considered "Lowest concentration." For laboratory parameters, the following were considered: hyperglycemia (fasting glucose $\geq$ 100 mg/dL); hypercholesterolemia (LDL-cholesterol $\geq$ 130 mg/dL); hypertriglyceridemia ($\geq$ 100 mg/dL for subjects aged 5 to 10 y; $\geq$ 150 mg/dL for subjects aged 11 to 19 y). For the anthropometric parameters, overweight was defined as the junction of overweight, obesity, and severe obesity, according to the Body Mass Index (BMI) for Age (BMI/A), using the child growth curves of the World Health Organization [34]. Cardiovascular risk was classified according to the Waist-to-Height Ratio (WHtR), being considered present when $\geq$ 0.5 [36].

People with more contact with the rural lifestyle tend to have a healthier intestinal gut microbiome, with a predominance of probiotic bacteria, including *Bifidobacterium* spp. [42]. In this study, there was no significant statistical difference in the *Bifidobacterium* spp. fecal abundance according to the geographic location of the population. However, the population was divided between rural and urban based on the school location rather than the place of residence. Most subjects who studied in rural schools also lived in this region, as did those who studied in urban schools. This may represent a limitation in interpreting this result since some of the evaluated subjects were in schools outside their residential region.

After childhood, the microbiota continues to develop, and the diet becomes primarily responsible for its structure, shape, and variety [43]. Due to the higher intake of fiber, plant-based diets are related to the more significant variation of microbial species, with *Firmicutes* and *Bacteroidetes* prevalence [44]. On the other hand, the dietary pattern rich in fats and animal protein is related to a greater abundance of bile-tolerant species, such as *Bacteroides* spp., and suppression of *Firmicutes* [45, 46]. Furthermore, the consumption of ultra-processed foods can, directly and indirectly, alter the composition of the gut microbiome due to changes in the density of micronutrients and energy, the presence of food additives, and Advanced Glycation End products from heat treatment during the processing of these foods [47].

The Brazilian population has a growing tendency to replace essential foods such as rice, beans, fruits, vegetables, beef, and milk with industrialized beverages and foods, such as cookies, processed meats, ready-to-eat foods, sugar, and salt. As a result, fruits and vegetable

consumption is lower among adolescents than among adults and the elderly. On the other hand, the consumption of ultra-processed foods is higher in this population [48]. In this study, it was observed that subjects who consumed more meat, compared to those who consumed less, had lower *Bifidobacterium* spp. fecal abundance. An important aspect that needs to be discussed is that, due to logistical issues, the food consumption assessment presented in this study was performed using a single 24-hour recall. For that, the given data represent food consumption on the day before the interview, not the usual pattern of consumption, and the interpretations derived from these findings should be made taking this limitation into account.

The main consequence of the gut microbiome imbalance is the increase of intestinal mucosa permeability, which can lead to endotoxemia by lipopolysaccharides synthesized by Gram-negative intestinal bacteria; increased synthesis of pro-inflammatory cytokines; macrophage infiltration into adipose tissue; and the consequent local and systemic inflammatory process, playing an essential role in triggering peripheral insulin resistance, obesity, metabolic syndrome and cardiovascular diseases [49]. Another important aspect is the microbiota's ability to metabolize choline and carnitine, a vitamin and amino acid, respectively, abundant in the Western dietary pattern because they are present in more significant amounts in animal foods. The result of these compounds' metabolism is the synthesis of trimethylamine, which is oxidized in the liver to trimethylamine-N-oxide. This compound suppresses enzymes synthesizing bile acids and cholesterol transporters, which is related to atherosclerosis pathogenesis [50]. In addition, the intestinal microbiota of individuals with cardiovascular diseases has a lower abundance of butyrate-producing strains, such as *Bifidobacterium* spp. [5].

In this study, the low *Bifidobacterium* spp. fecal abundance was associated with a higher prevalence of hyperglycemia, and a higher abundance of this bacterial genus in the intestinal microbiota was associated with cardiovascular protection. Using the prevalence ratio in this context offers a lucid comprehension of the way in which alterations in the exposure (*Bifidobacterium* spp. abundance) correspond to modifications in the outcome (hyperglycemia and cardiovascular risk). This metric proves especially pertinent when dealing with prevalent outcomes and cross-sectional study designs, as it directly quantifies the association's potency by gauging shifts in prevalence. Further studies are suggested to investigate the influence of physical activity on these findings.

## Conclusions

Our findings strongly suggest that *Bifidobacterium* spp. fecal abundance is related to dietary and anthropometric parameters in school-aged subjects, and its increase is associated with positive metabolic outcomes. These data reinforce the need for public policies on Food and Nutrition Security to prevent NCDs among children and adolescents.

## Acknowledgments

Special thanks to the Public Ministry of Bahia and the Health and Education Departments of the municipalities involved for enabling the logistics for data collection; as well as the nutritionists responsible for the National School Feeding Program in the cities; to Nutrition students Bruna Cerqueira and Bruno Cruz, who voluntarily contributed to data collection; and to the public and private laboratories that supported the collection and analysis of laboratory indicators.

## Author Contributions

**Conceptualization:** Camilla A. Menezes, Dalila L. Zanette, Ana Marice T. Ladeia, Nelzair A. Vianna, Ricardo R. Oliveira.

**Data curation:** Camilla A. Menezes, Dalila L. Zanette, Letícia B. Magalhães, Jacqueline Tereza da Silva, Renata M. R. S. Lago, Alexvon N. Gomes, Ronald A. dos Santos.

**Formal analysis:** Camilla A. Menezes, Dalila L. Zanette, Letícia B. Magalhães, Jacqueline Tereza da Silva.

**Investigation:** Camilla A. Menezes, Dalila L. Zanette, Letícia B. Magalhães, Jacqueline Tereza da Silva, Renata M. R. S. Lago, Alexvon N. Gomes, Ronald A. dos Santos, Ana Marice T. Ladeia, Nelzair A. Vianna, Ricardo R. Oliveira.

**Methodology:** Camilla A. Menezes, Dalila L. Zanette, Ana Marice T. Ladeia, Nelzair A. Vianna, Ricardo R. Oliveira.

**Project administration:** Ana Marice T. Ladeia, Nelzair A. Vianna, Ricardo R. Oliveira.

**Supervision:** Dalila L. Zanette, Ana Marice T. Ladeia, Nelzair A. Vianna, Ricardo R. Oliveira.

**Validation:** Camilla A. Menezes.

**Visualization:** Ricardo R. Oliveira.

**Writing – original draft:** Camilla A. Menezes.

**Writing – review & editing:** Jacqueline Tereza da Silva, Renata M. R. S. Lago, Alexvon N. Gomes, Ronald A. dos Santos, Ana Marice T. Ladeia, Nelzair A. Vianna, Ricardo R. Oliveira.

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
