## [Editor Report · Decision Letter 0]

15 Dec 2022

PONE-D-22-33730Higher Bifidobacterium spp. fecal abundance is associated with a lower prevalence of hyperglycemia and cardiovascular risk markers among schoolchildren from Bahia, BrazilPLOS ONE

Dear Dr. Oliveira,

Thank you for submitting your manuscript to PLOS ONE. After careful consideration, we feel that it has merit but does not fully meet PLOS ONE’s publication criteria as it currently stands. Therefore, we invite you to submit a revised version of the manuscript that addresses the points raised during the review process.

ACADEMIC EDITOR: The work brings important considerations about the nutritional conditions of Brazilian children, presenting a promising health marker. However, it lacks English revision, and that the tables are clearer, presenting results from only one type of statistical analysis.==============================

We look forward to receiving your revised manuscript.

Kind regards,

Leonardo Costa Pereira, Doctor

Academic Editor

PLOS ONE
---

## [Author Response · Author response to Decision Letter 0]

23 Jan 2023

Dear Academic Editor and Reviewers, 

We are thankful for the time and attention dedicated to this first review of our manuscript “Higher Bifidobacterium spp. fecal abundance is associated with a lower prevalence of hyperglycemia and cardiovascular risk markers among schoolchildren from Bahia, Brazil.” Each point raised by the review is explored as follows. 

Editor comments:

Point 1: The work brings important considerations about the nutritional conditions of Brazilian children, presenting a promising health marker. However, it lacks English revision, and that the tables are clearer, presenting results from only one type of statistical analysis.

Response 1: Thank you for this positive feedback. The manuscript underwent another English revision. Regarding the tables, unfortunately, it is not possible to standardize the statistical analysis because some of the numerical variables have two groups and others have three or more groups, which leads to apply different statistical tests to compare them. 

Journal Requirements:

Point 1: Please ensure that your manuscript meets PLOS ONE's style requirements, including those for file naming.

Response 1: We double-checked the style requirements and hope that the revised manuscript submitted achieves them. 

Point 2: We note that the grant information you provided in the ‘Funding Information’ and ‘Financial Disclosure’ sections do not match.

Response 2: During the manuscript’s writing, the first author received a research grant from the Foundation for Research Support of the State of Bahia - Fundação de Amparo à Pesquisa do Estado da Bahia (FAPESB, Brazil). We intend to standardize this information during the resubmission process. 

Point 3: In your Data Availability statement, you have not specified where the minimal data set underlying the results described in your manuscript can be found. PLOS defines a study's minimal data set as the underlying data used to reach the conclusions drawn in the manuscript and any additional data required to replicate the reported study findings in their entirety. All PLOS journals require that the minimal data set be made fully available.

Response 3: The dataset used to support the results presented in the manuscript is in process of upload at the FIOCRUZ institutional data repository (https://dadosdepesquisa.fiocruz.br/; ARCA DADOS). A specific DOI number will be attributed to the dataset and will be informed prior to the publication.

Point 4: PLOS requires an ORCID iD for the corresponding author in Editorial Manager on papers submitted after December 6th, 2016. Please ensure that you have an ORCID iD and that it is validated in Editorial Manager. To do this, go to ‘Update my Information’ (in the upper left-hand corner of the main menu), and click on the Fetch/Validate link next to the ORCID field. This will take you to the ORCID site and allow you to create a new iD or authenticate a pre-existing iD in Editorial Manager.

Response 4: The corresponding author has an Editorial Manager validated ORCID iD and the information was updated. 

Point 5: While revising your submission, please upload your figure files to the Preflight Analysis and Conversion Engine (PACE) digital diagnostic tool, https://pacev2.apexcovantage.com/.

Response 5: We used the PACE digital diagnostic tool, and the figures now attend to the PLOS One requirements. 

We hope these changes attend to the points raised by the first Academic Editor review.

Best regards, 

Ricardo Riccio Oliveira

Instituto Gonçalo Moniz, Fundação Oswaldo Cruz (Fiocruz)

+55 71 3176-2347 | +55 71 3176-2266

ricardo.riccio@fiocruz.br

---

## [Decision Letter · Decision Letter 1]

10 Apr 2023

PONE-D-22-33730R1Higher Bifidobacterium spp. fecal abundance is associated with a lower prevalence of hyperglycemia and cardiovascular risk markers among schoolchildren from Bahia, BrazilPLOS ONE

Dear Dr. Oliveira,

Thank you for submitting your manuscript to PLOS ONE. After careful consideration, we feel that it has merit but does not fully meet PLOS ONE’s publication criteria as it currently stands. Therefore, we invite you to submit a revised version of the manuscript that addresses the points raised during the review process.

There remain some important issues that the authors need to address more adequately.

We look forward to receiving your revised manuscript.

Kind regards,

Brenda A Wilson, Ph.D.

Academic Editor

PLOS ONE

Note: HTML markup is below. Please do not edit.]

Reviewers' comments:

Reviewer's Responses to Questions

**Comments to the Author**

1. If the authors have adequately addressed your comments raised in a previous round of review and you feel that this manuscript is now acceptable for publication, you may indicate that here to bypass the “Comments to the Author” section, enter your conflict of interest statement in the “Confidential to Editor” section, and submit your "Accept" recommendation.

Reviewer #1: (No Response)

Reviewer #2: All comments have been addressed

2. Is the manuscript technically sound, and do the data support the conclusions?

Reviewer #1: Partly

Reviewer #2: Yes

3. Has the statistical analysis been performed appropriately and rigorously? 

Reviewer #1: I Don't Know

Reviewer #2: Yes

4. Have the authors made all data underlying the findings in their manuscript fully available?

Reviewer #1: Yes

Reviewer #2: Yes

5. Is the manuscript presented in an intelligible fashion and written in standard English?

Reviewer #1: Yes

Reviewer #2: Yes

6. Review Comments to the Author

Reviewer #1: This is a very interesting manuscript.

I have a few questions or suggestions:

1. The first paragraph in Discussion would probably fit better to Introduction (incorporated to the current Introduction).

2. In the paragraph starting on line 320: any previous information in literature how diet would modify the Bifidobacterium ssp. in the gut microbiota after breastfeeding stopped?

3. Who carried out the 24-hr recall? Were they trained to carry out the recall (that can be quite challenging if the interviewer does not know for example the local food supply well, and/or not mastering the serving size estimation and conversion to weight units)? How long did children fast before the blood draw?

4. How was the food consumption information from 24-hr recall forms and food composition tables converted into data on dietary intake (energy and macronutrients) of a child/teenager: did you use a specific in-house program or an application for it? One-time 24-hr recall is not ideal in studying dietary intake on individual level because diet can vary a lot from day to day. Not very strong conclusions can be made based on dietary intake (one-time 24-hr recall).

5. There were several dietary biomarkers measured from the blood samples like ferritin, vitamins B12 and D. The results are missing. If you don’t want to report them please do not mention them in the methods.

6. Any adjustments in the statistical analyses? Age could be a confounder.

7. Table 1. You indicate that 179 children were breastfed at infancy. However, if you sum up the N’s in two categories of BF durations (108+79) you get 187 children. Where is the difference coming from?

8. It would be clearer if you first give the number of children breastfed <6 months (n=79) and then those breastfed >6 months (n=108). I think “>6 months” is clearer than “Up to sixth month” and “<6 months” instead of “Less than sixth month”. This is a matter of taste of course.

9. It says in line 196 “exclusive breastfeeding” but should probably be any breastfeeding or just ‘breastfeeding’ to match with Table 1. Hard to believe that 58% of the children were exclusively breastfed by age of 6 months or longer (no formula or solid foods). This is probably referring to any breastfeeding.

10. In Table 2 I would say “Energy (kcal)”.

11. Please use Fats instead of Lipids.

12. In the foot note or somewhere else there should be a brief definition of “processed foods” and of “ultra-processed foods” so that the reader understands what is the difference between these two types of foods.

13. Please check the use of comma and period in the text. Now it is not consistent.

14. Studying the topic further including physical activity sounds very good! All the best for it!

Reviewer #2: Thank you for the opportunity to review this manuscript titled “Higher Bifidobacterium spp. fecal abundance is associated with a lower prevalence of hyperglycemia and cardiovascular risk markers among schoolchildren from Bahia, Brazil”. The authors conducted a cross sectional study aimed to evaluate the fecal abundance of BIF in a group of schoolchildren from public schools in Bahia, Brazil, and to investigate their relationship with food consumption and laboratory and anthropometric characteristics. The authors found that low BIF abundance was associated with a higher prevalence of hyperglycemia and high WHtR. While the manuscript provides a detailed description of the data source, questionnaires used to measure different variables, and results, I have a few comments for the authors to address. Additionally, I suggest that the authors improve their written English throughout the manuscript.

Line 89: The authors should explain the rationale for only selecting individuals aged 5 and 19 years in the study.

Line 90: The authors should briefly explain the rationale for excluding patients with “previous diagnosis of food allergies and intolerances and the use of antibiotics 30 days before the faecal material collection.”

Line 97: The authors mention they used a previously structured questionnaire for clinical and demographic assessment. However, references are not provided.

Line 104: Please provide a reference for the 24-hour recall instrument used to measure food consumption.

Line 180: The authors used logistic regression model to generate prevalence ratio for the outcomes. However, the correct statistical term should be “Odds ratio” and not “prevalence ratio”. Please correct throughout the manuscript.

Line 181: How did the authors handle missing values when the responder did not respond to a specific question? Was the data imputed?

Line 181: The authors should report findings of multicollinearity in the logistic regression model.

Lines 265: In Table 4, 95% CI is reported for each variable of interest. There is a typo in one of the columns i.e.,” CL95%” should be corrected to “95%CI”.

Lines 265: The authors should report the findings of the multivariate logistic regression model.

Line 342: The authors should list all the limitations of a cross sectional study including but not limited to “selection bias, measurement bias, confounding, limited generalizability, recall bias etc.”. The authors should consider these limitations when interpreting the results.

7. PLOS authors have the option to publish the peer review history of their article (what does this mean?). If published, this will include your full peer review and any attached files.

Reviewer #1: No

Reviewer #2: No

---

## [Author Response · Author response to Decision Letter 1]

25 May 2023

Reviewer 1

Comment: This is a very interesting manuscript.

Response: Thank you for recognizing our efforts in conducting this research. 

Point 1: The first paragraph in Discussion would probably fit better to Introduction (incorporated to the current Introduction). 

Response 1: We agree with that suggestion. The information from this paragraph was incorporated into the first and last paragraphs of the introduction.

Point 2: In the paragraph starting on line 320: any previous information in literature how diet would modify the Bifidobacterium ssp. in the gut microbiota after breastfeeding stopped?

Response 2: According to the literature, as human milk is a natural source of bifidobacteria, once breastfeeding is discontinued or complementary feeding is initiated, there is a reduction in the availability of this bacterial genus to colonize the intestine. Simultaneously, the introduction of other foods, and therefore other energy substrates, promotes the growth of other bacterial genera (10.1016/j.amjms.2018.08.005). This information was inserted in the second paragraph of the discussion.

Point 3: Who carried out the 24-hr recall? Were they trained to carry out the recall (that can be quite challenging if the interviewer does not know for example the local food supply well, and/or not mastering the serving size estimation and conversion to weight units)? How long did children fast before the blood draw? 

Response 3: The 24-hour recall was conducted by the nutritionists in our research group, who were trained for this task, as we recognize the need for careful attention in the approach, recording, and analysis process. These nutritionists had prior access to the menus served in schools, which had been the subject of another publication by our group, and this was crucial in understanding the local dietary habits (10.3390/nu14071519). To assist in the interviews, a photographic manual of typical regional foods was used, displaying their usual portion sizes. The images were accompanied by codes that were used to convert the portions into standardized measurement units (grams or milliliters). Participants fasted for 8 hours before blood collection. These pieces of information were inserted in the "Food consumption evaluation" and “Blood and stool assessment” subsection of the "Materials and Methods" section. 

Point 4: How was the food consumption information from 24-hr recall forms and food composition tables converted into data on dietary intake (energy and macronutrients) of a child/teenager: did you use a specific in-house program or an application for it? One-time 24-hr recall is not ideal in studying dietary intake on individual level because diet can vary a lot from day to day. Not very strong conclusions can be made based on dietary intake (one-time 24-hr recall).

Response 4: The 24-hour recalls recorded the consumed foods and portions. We inputted the consumption data of each participant into an Excel spreadsheet. Afterward, we accessed the Brazilian Food Composition Table database, also in Excel format. Using R software, we cross-referenced the information to identify the number of calories, macronutrients, and micronutrients consumed by each participant. For foods not available in that table, we used the Centesimal Food Composition Table of the Brazilian Institute of Geography and Statistics. This information was inserted in the "Food consumption evaluation" subsection of the "Materials and Methods" section. We agree with the observation regarding the assessment of dietary intake based on a single 24-hour recall. Unfortunately, due to logistical constraints, it was not possible to conduct the assessment on multiple occasions. Aware of this limitation, we emphasize and acknowledge this in the discussion, in the sixth paragraph. However, considering the novelty of the results in our context, we deemed it important to share them with the scientific community. 

Point 5: There were several dietary biomarkers measured from the blood samples like ferritin, vitamins B12 and D. The results are missing. If you don’t want to report them please do not mention them in the methods. 

Response 5: We appreciate this observation. Indeed, these data were used in another publication (10.3390/nu15020381), but since they are not explored in the analysis of this study, they should not be included in the methods section. We have removed this information from the methodological description. 

Point 6: Any adjustments in the statistical analyses? Age could be a confounder. 

Response 6: We tested whether certain variables, such as gender, age, and population location (rural or urban), could act as confounders, but the result was negative.

Point 7: Table 1. You indicate that 179 children were breastfed at infancy. However, if you sum up the N’s in two categories of BF durations (108+79) you get 187 children. Where is the difference coming from?

Response 7: Thank you for bringing attention to this typing error. 173 participants reported being breastfed, of which 100 (not 108) were exclusively breastfed until the sixth month of life, and 73 (not 79) initiated complementary feeding before that period. We have made the corrections in Table 1.

Point 8: It would be clearer if you first give the number of children breastfed <6 months (n=79) and then those breastfed >6 months (n=108). I think “>6 months” is clearer than “Up to sixth month” and “<6 months” instead of “Less than sixth month”. This is a matter of taste of course.

Response 8: Regarding the first observation, which suggests describing first the number of children who were not breastfed until the sixth month of life, followed by the number of children breastfed until that period, we have chosen to maintain the order as described in the table, as the remaining variables follow the logic of describing what would be "indicated" first and then what would be unfavorable. As for the second observation, we agree with your suggestion. It was a literal translation from Portuguese to English, but it is better understood the way you have suggested. We have altered Table 1. 

Point 9: It says in line 196 “exclusive breastfeeding” but should probably be any breastfeeding or just ‘breastfeeding’ to match with Table 1. Hard to believe that 58% of the children were exclusively breastfed by age of 6 months or longer (no formula or solid foods). This is probably referring to any breastfeeding. 

Response 9: In this case, it is indeed about exclusive breastfeeding, without the introduction of other foods or formulas. This question was posed to parents or guardians during the interview, explaining the meaning of exclusive breastfeeding, and 58% of them responded affirmatively. We included this information in the table to confirm the data. This finding supports information from the Brazilian Ministry of Health, which estimates the population rate at 45.8% and assumes that this practice is more prevalent in populations from inland cities and rural areas. (https://aps.saude.gov.br/noticia/18257#:~:text=A%20amamenta%C3%A7%C3%A3o%20exclusiva%2C%20no%20Brasil,meses%20de%20vida%20at%C3%A9%202025.). 

Point 10: In Table 2 I would say “Energy (kcal)”. 

Response 10: We agree and made the change in Table 2. 

Point 11: Please use Fats instead of Lipids. 

Response 11: We agree and made the change in Table 2. 

Point 12: In the foot note or somewhere else there should be a brief definition of “processed foods” and of “ultra-processed foods” so that the reader understands what is the difference between these two types of foods. 

Response 12: We agree and included this information in the caption of Table 2.

Point 13: Please check the use of comma and period in the text. Now it is not consistent. 

Response 13: Thank you for this feedback. We double-checked this. 

Point 14: Studying the topic further including physical activity sounds very good! All the best for it! 

Response 14: Thank you for this positive feedback. 

Reviewer 2

Comment: Thank you for the opportunity to review this manuscript titled “Higher Bifidobacterium spp. fecal abundance is associated with a lower prevalence of hyperglycemia and cardiovascular risk markers among schoolchildren from Bahia, Brazil”. The authors conducted a cross sectional study aimed to evaluate the fecal abundance of BIF in a group of schoolchildren from public schools in Bahia, Brazil, and to investigate their relationship with food consumption and laboratory and anthropometric characteristics. The authors found that low BIF abundance was associated with a higher prevalence of hyperglycemia and high WHtR. While the manuscript provides a detailed description of the data source, questionnaires used to measure different variables, and results, I have a few comments for the authors to address. Additionally, I suggest that the authors improve their written English throughout the manuscript.

Response: We appreciate your review, comments, and contributions. Regarding improving written English throughout the manuscript, we have double-checked it again.

Point 1: Line 89: The authors should explain the rationale for only selecting individuals aged 5 and 19 years in the study. 

Response 1: We included individuals aged 5 to 19 years because most school-age children in the region fall within this age range. This information was inserted in the "Population and Data Collection" subsection of the "Materials and Methods" section.

Point 2: Line 90: The authors should briefly explain the rationale for excluding patients with “previous diagnosis of food allergies and intolerances and the use of antibiotics 30 days before the faecal material collection.” 

Response 2: We excluded individuals with food allergies and intolerances and individuals who had used antibiotics within 30 days before stool collection because these conditions alter the gut microbiota. This information was inserted in the "Population and Data Collection" subsection of the "Materials and Methods" section.

Point 3: Line 97: The authors mention they used a previously structured questionnaire for clinical and demographic assessment. However, references are not provided. 

Response 3: This was a questionnaire developed by the research group exclusively for use in this study. The intention in mentioning that this assessment was conducted through an interview guided by a questionnaire was to demonstrate standardization and ensure that all participants were asked the same questions. That is why we did not consider it possible to reference the document.

Point 4: Line 104: Please provide a reference for the 24-hour recall instrument used to measure food consumption. 

Response 4: The 24-hour recall is a strategy for collecting dietary intake data (retrospective, quantitative - foods consumed on the day before the interview) and not a standardized or published document format. In the case of retrospective qualitative assessments, such as food frequency questionnaires, there are validated standards in the literature, but for the 24-hour recall, such standards do not exist. Therefore, we cannot provide a reference for the instrument in this case. In any case, this instrument was used in a publication by the research group, which is also a result of the current research (10.3390/nu15020381).

Point 5: Line 180: The authors used logistic regression model to generate prevalence ratio for the outcomes. However, the correct statistical term should be “Odds ratio” and not “prevalence ratio”. Please correct throughout the manuscript. 

Response 5: According to the literature that we used to support our statistical analysis, although the Odds Ratio (OR) or the Prevalence Ratio (PR) could be good estimators of the association between a dichotomous dependent variable and one or more independent variables, traditionally most studies have used OR, calculated with logistic regression, to estimate the association. However, although OR is a good estimator of PR when the prevalence is low, it is known that OR overestimates PR when the prevalence is moderate or high. We understand that, as this is a cross-sectional study, the appropriate analysis and term would be "Prevalence Ratio" rather than "Odds Ratio," which we understand is typically used in case-control studies (10.20882/adicciones.823). Based on that we have opted to keep PR in the manuscript. 

Point 6: Line 181: How did the authors handle missing values when the responder did not respond to a specific question? Was the data imputed? 

Response 6: The missing data were not imputed. For the variables for which we couldn't obtain data, the missing data was subtracted from the total study sample (n=190), and the remaining data were analyzed.

Point 7: Line 181: The authors should report findings of multicollinearity in the logistic regression model. 

Response 7: We observed through Spearman's correlation that only one dependent variable (WHtR) showed a statistically significant relationship with the tested independent variable (fecal Bifidobacterium spp. concentration), but this relationship was weak. Similarly, as only one independent variable was analyzed, there was no correlation with any other, and thus we did not observe multicollinearity in the logistic regression model used.

Point 8: Lines 265: In Table 4, 95% CI is reported for each variable of interest. There is a typo in one of the columns i.e.,” CL95%” should be corrected to “95%CI”. 

Response 8: Thank you for the observation. We have corrected the error.

Point 9: Lines 265: The authors should report the findings of the multivariate logistic regression model. 

Response 9: As the variables gender, age, and population location did not act as confounders, we did not use multivariate analysis.

Point 10: Line 342: The authors should list all the limitations of a cross sectional study including but not limited to “selection bias, measurement bias, confounding, limited generalizability, recall bias etc.”. The authors should consider these limitations when interpreting the results. 

Response 10: Understanding that the variables "delivery type, breastfeeding, and use of antibiotic therapy in childhood" had responses obtained from information provided by the participant's caregiver, rather than extracted from medical records or other documents, it is possible that it represents a measurement bias. Therefore, we described the limitation of interpreting the data resulting from these variables in the third paragraph of the discussion. Similarly, the variable "population location" actually refers to the location of the school where the participant studied, not their place of residence, which may also introduce selection bias. This limitation was addressed in the fourth paragraph of the discussion. Lastly, the application of the 24-hour recall for assessing dietary intake at only one point in time may introduce bias. Therefore, in the study, dietary intake was interpreted as "the day before the interview" rather than reflecting the habitual consumption pattern of the population. This limitation was explained in the sixth paragraph. Furthermore, the study's conclusions were based on the results found, without intending to generalize to populations of different age groups or contexts, nor to other bacterial genera that comprise the gut microbiota.

---

## [Decision Letter · Decision Letter 2]

29 Jun 2023

PONE-D-22-33730R2Higher Bifidobacterium spp. fecal abundance is associated with a lower prevalence of hyperglycemia and cardiovascular risk markers among schoolchildren from Bahia, BrazilPLOS ONE

Dear Dr. Oliveira,

Thank you for submitting your manuscript to PLOS ONE. After careful consideration, we feel that it has merit but does not fully meet PLOS ONE’s publication criteria as it currently stands. Therefore, we invite you to submit a revised version of the manuscript that addresses the points raised during the review process. Both of the reviewers felt that the authors were adequately responsive to previous reviewers' suggestions. However, there are still some concerns that need to be fully addressed. Both felt the writing needs improvement. In particular, the authors should address the major missing element noted by one of the reviewers, namely discussion about the current state of knowledge in the field, with appropriate representative studies, background, and references in the Introduction section and followup in the Discussion section. 

We look forward to receiving your revised manuscript.

Kind regards,

Brenda A Wilson, Ph.D.

Academic Editor

PLOS ONE

Journal Requirements:

Additional Editor Comments:

No additional experiments need to be done, but the writing needs improvement.

Reviewers' comments:

Reviewer's Responses to Questions

**Comments to the Author**

1. If the authors have adequately addressed your comments raised in a previous round of review and you feel that this manuscript is now acceptable for publication, you may indicate that here to bypass the “Comments to the Author” section, enter your conflict of interest statement in the “Confidential to Editor” section, and submit your "Accept" recommendation.

Reviewer #3: All comments have been addressed

Reviewer #4: All comments have been addressed

2. Is the manuscript technically sound, and do the data support the conclusions?

Reviewer #3: Yes

Reviewer #4: Yes

3. Has the statistical analysis been performed appropriately and rigorously? 

Reviewer #3: Yes

Reviewer #4: Yes

4. Have the authors made all data underlying the findings in their manuscript fully available?

Reviewer #3: Yes

Reviewer #4: Yes

5. Is the manuscript presented in an intelligible fashion and written in standard English?

Reviewer #3: Yes

Reviewer #4: No

6. Review Comments to the Author

Reviewer #3: The manuscript is a good piece of work

The author has done justice to the work in my view

the minor revisions are advised to address by the author it can be addressed further for publication

Reviewer #4: The authors performed a cross-sectional analysis of the relationship between the fecal abundance of Bifidobacterium spp. with dietary and anthropometric parameters among the school-aged population. The analyses linked positive metabolic outcomes with increased Bifidobacterium spp. fecal abundance. The data provide a framework to further understand the effects of Bifidobacterium spp. on children's metabolic outcomes. The authors were responsive to reviewers' suggestions; most revisions and select omissions are justified. However, there are still some concerns that need to be fully addressed.

One major missing element is the current state of knowledge about the field. The authors should accurately represent appropriate studies and references in the Introduction section. The current version is not cohesive and lacks sufficient background, including but not limited to:

- Line 73-75: missing valuable reference for the statement about “conflicting results”. The previous studies about the relationship between Bifidobacterium spp. and metabolic outcomes in the child population should be referenced for context

- Line 70: The authors should double-check the number. According to reference 3 and current NCBI Taxonomy ID 1678, there are more than 29 Bifidobacterium species.

- Line 62: The evidence of “tropism” cannot be found in reference 2. Based on reference 2, one of the significant functions of gut microbiota is controlling epithelial cell proliferation and differentiation, but “tropism” was not mentioned in reference 2.

7. PLOS authors have the option to publish the peer review history of their article (what does this mean?). If published, this will include your full peer review and any attached files.

Reviewer #3: No

Reviewer #4: No

---

## [Author Response · Author response to Decision Letter 2]

14 Aug 2023

The points raised by the reviewers are explored as follows. 

Reviewer 3

Comment 1: The work is indeed a good piece of work and shows the relevance in current context with regard to food intake and gut flora behaviors string response in research field. 

Response: We are grateful for your positive feedback on our work. It's encouraging to hear that you find the relevance of our study in the current context of food intake and its impact on the gut microbiome. We will continue to strive for excellence in our research endeavors. 

Comment 2: The early reviewers (both) have addressed major questions and the author also tried with its maximum effort to address the queries with responsibility.

Response: Thank you for recognizing our commitment to addressing the reviewers' concerns. We have indeed put forth our utmost effort to respond to the queries responsibly. Your acknowledgment is truly motivating as we aim to ensure the quality and rigor of our work. 

Comment 3: I have only to say about the discussion part where statistical interpretation if possible can be addressed.

Response: We appreciate your suggestion regarding the discussion section and the inclusion of a statistical interpretation. We have considered your input and worked towards enhancing the depth of analysis in that part of the paper and included a sentence in the last paragraph of the section.

Comment 4: Rest I believe the manuscript is good to consider for publication.

Response: Thank you for considering our manuscript favorably. Your confidence in the quality of our work is truly motivating. We will ensure that any remaining aspects are polished to meet the standards for publication. 

Reviewer 4

Comment 1: The authors performed a cross-sectional analysis of the relationship between the fecal abundance of Bifidobacterium spp. with dietary and anthropometric parameters among the school-aged population. The analyses linked positive metabolic outcomes with increased Bifidobacterium spp. fecal abundance. The data provide a framework to further understand the effects of Bifidobacterium spp. on children's metabolic outcomes. The authors were responsive to reviewers' suggestions; most revisions and select omissions are justified. However, there are still some concerns that need to be fully addressed.

Response: We sincerely appreciate your thorough review of our manuscript and your acknowledgment of our efforts to address previous reviewers' suggestions. It's encouraging to know that most of the revisions and omissions were justified and well-received. We value your feedback on the remaining concerns, and we are committed to fully addressing them to ensure the overall quality of the manuscript.

Comment 2: One major missing element is the current state of knowledge about the field. The authors should accurately represent appropriate studies and references in the Introduction section.

Response: Your observation about the need to include a comprehensive overview of the current state of knowledge in the field is invaluable. We have diligently worked on enhancing the Introduction section by accurately representing relevant studies and incorporating appropriate references. Your guidance will undoubtedly contribute to strengthening the overall context and significance of our work. 

Point 1: The current version is not cohesive and lacks sufficient background, including but not limited to: - Line 73-75: missing valuable reference for the statement about “conflicting results”. The previous studies about the relationship between Bifidobacterium spp. and metabolic outcomes in the child population should be referenced for context. 

Response: We acknowledge the need for better cohesion and background in the current version. We addressed the missing reference and included relevant studies on the relationship between Bifidobacterium spp. and metabolic outcomes, including in children, to provide the necessary context (references 16 to 19 - Michels N. et al, 2022; Gholizadeh P. et al, 2019; Guo Q. et al, 2019; Chen AC. et al, 2022). 

Point 2: Line 70: The authors should double-check the number. According to reference 3 and current NCBI Taxonomy ID 1678, there are more than 29 Bifidobacterium species. 

Response: Thank you for bringing this to our attention. We have carefully reviewed the reference (Hidalgo-Cantabrana C. et al, 2017) and found that there are indeed more than 50 documented species of Bifidobacterium. We have updated our information accordingly in the first paragraph of page 3.

Point 3: Line 62: The evidence of “tropism” cannot be found in reference 2. Based on reference 2, one of the significant functions of gut microbiota is controlling epithelial cell proliferation and differentiation, but “tropism” was not mentioned in reference 2. 

Response: The text mentions that the trophic function of the intestinal microbiota occurs through the stimulation of proliferation and differentiation of the intestinal epithelium, as well as the development and modulation of the immune system. This information was described in previous reference 1 (Beserra BTS, 2014 – the first paragraph of page 31) and not in the previous reference 2 (Gomaa EZ, 2020). Nevertheless, the suggestion to provide new references to better contextualize the topic has been addressed, and we have included the following: Manos J, 2022; Liang X. et al, 2022; Krakovski MA. et al, 2022; Zhang W. et al, 2022; Aleman RS, Moncada M, Aryana KJ, 2023; Dowling LR. et al, 2022 (references 6 to 11).

---

## [Editor Report · Decision Letter 3]

17 Aug 2023

Higher Bifidobacterium spp. fecal abundance is associated with a lower prevalence of hyperglycemia and cardiovascular risk markers among schoolchildren from Bahia, Brazil

PONE-D-22-33730R3

Dear Dr. Oliveira,

We’re pleased to inform you that your manuscript has been judged scientifically suitable for publication and will be formally accepted for publication once it meets all outstanding technical requirements.

Kind regards,

Brenda A Wilson, Ph.D.

Academic Editor

PLOS ONE
---

## [Editor Report · Acceptance letter]

10 Oct 2023

PONE-D-22-33730R3 

Higher *Bifidobacterium* spp. fecal abundance is associated with a lower prevalence of hyperglycemia and cardiovascular risk markers among schoolchildren from Bahia, Brazil 

Dear Dr. Oliveira:

I'm pleased to inform you that your manuscript has been deemed suitable for publication in PLOS ONE. Congratulations! Your manuscript is now with our production department. 

Kind regards, 

on behalf of

Dr. Brenda A Wilson 

Academic Editor

PLOS ONE